# Prevalence and Risk Factors Associated with Feline Infectious Peritonitis (FIP) in Mainland China between 2008 and 2023: A Systematic Review and Meta-Analysis

**DOI:** 10.3390/ani14081220

**Published:** 2024-04-18

**Authors:** Tingyu Hu, Huiling Zhang, Xueping Zhang, Xingping Hong, Tangjie Zhang

**Affiliations:** 1Institute of Comparative Medicine, College of Veterinary Medicine, Yangzhou University, Yangzhou 225009, China; hty201804107@foxmail.com (T.H.); zhangxp1995@foxmail.com (X.Z.); xingpinghong@foxmail.com (X.H.); 2Jiangsu Co-Innovation Center for Prevention and Control of Important Animal Infectious Diseases and Zoonoses, Yangzhou 225009, China; 3Independent Researcher, New York, NY 11355, USA; yzzhanghl@hotmail.com

**Keywords:** feline infectious peritonitis, meta-analysis, mainland China, cats

## Abstract

**Simple Summary:**

While it is widely believed that feline infectious peritonitis (FIP) is a dangerous epidemic disease, the prevalence of FIP in Chinese cats is currently unknown. Our findings indicate that the prevalence of FIP in Chinese cats is influenced by age, gender, and breed. We recommend controlling the scale of cat breeding, enhancing cat immunity, and particularly focusing on health management in kittens and continuous monitoring of FIP in cats.

**Abstract:**

To evaluate the overall prevalence of FIP infection in cats in mainland China and associated risk factors, studies on the prevalence of FIP conducted from 1 January 2008 to 20 December 2023 were retrieved from five databases—CNKI, Wanfang, PubMed, Web of Science, and ScienceDirect—and comprehensively reviewed. The 21 studies selected, with a total of 181,014 samples, underwent a rigorous meta-analysis after quality assessment. The results revealed a 2% prevalence of FIP (95% CI: 1–2%) through the random-effects model, showing considerable heterogeneity (I^2^ = 95.2%). The subsequent subgroup analysis revealed that the age and gender of cats are significant risk factors for FIP infection in mainland China. In order to effectively reduce and control the prevalence of FIP on the Chinese mainland, we suggest improving the immunity of cats, with special attention given to health management in kittens and intact cats, and continuously monitoring FIPV.

## 1. Introduction

Feline infectious peritonitis (FIP) is a progressive systemic disease characterized by a wide range of clinical symptoms, represented by peritonitis, the accumulation of a large amount of ascites, and a high mortality rate [1]. It is highly contagious and spreads efficiently through fecal–oral transmission, leading to a high prevalence in multi-cat environments such as breeding catteries [2], shelter or rescue facilities, and situations involving animal hoarding [3]. FIP, a viral-induced immune-mediated disease with a high fatality rate, is caused by the feline infectious peritonitis virus (FIPV). FIPV is a member of the genus Alphacoronavirus, a group of enveloped, single-stranded, positive-sense RNA viruses, stemming from a mutation in the feline coronavirus (FCoV) [4,5]. FIP was first identified as a distinct disease in 1963 by Dr. Jean Holzworth et al. at the Angell Memorial Animal Hospital in Boston, USA [6]. Feline coronavirus (FCoV) is prevalent worldwide and is commonly found in most cat populations [3,7]. Studies indicate that at least 50% of cats in the United States and Europe possess antibodies against coronaviruses. In Switzerland, 80% of breeding cats and 50% of free-roaming cats tested positive for antibodies. Similarly, in Great Britain, 82% of show cats, 53% of cats in breeding institutions, and 15% of cats in single-cat households were found to have antibodies [7,8]. FIP poses a significant threat to feline health. The prevalence of coronavirus infection is notably high in cats, ranging from 25% to 40% in domestic pets and escalating to 80% and 100% in multi-cat households or settlements [9]. FCoV consists of two distinct pathotypes known as feline enteric coronavirus (FECV) and feline infectious peritonitis virus (FIPV), each inducing different pathological symptoms [10]. While FCoV can cause benign enteral infections, it may also lead to FIP. FCoV-positive cats often progress to feline enterocoronavirus (FECV) and FIPV, with 5% to 12% developing FIP [11]. Normal cats may carry viruses such as FECV or FCoV, but immunocompromised cats have a higher likelihood of virus mutation and FIPV development [12]. The mutations and pathogenic mechanisms responsible for the development of feline infectious peritonitis (FIP) remain unknown [13,14,15]. FIP accounted for an estimated 0.3% to 1.4% of feline deaths in veterinary institutions [16]. While domestic cats are susceptible, wild felids also experience a certain incidence rate and high mortality [17,18]. Diagnosing FIP is relatively straightforward when a cat with typical characteristics presents with effusion. However, in the absence of effusion, diagnosis becomes challenging due to the varied and non-specific clinical signs. Traditionally, the definitive diagnosis of FIP relies on histopathological examination of the tissue [19,20,21].

Historically, three primary pharmacological approaches have been utilized to treat FIP: (1) immunomodulators to non-specifically stimulate the patient’s immune system, aiming to diminish the clinical effects of the virus through triggering a robust immune response; (2) immunosuppressive agents to temporarily alleviate clinical signs; and (3) repurposed human antiviral drugs. However, none of these approaches have been proven to be reliably effective treatment options for FIPV. Since 2016, antiviral drugs that effectively inhibit the replication of FCoV have been developed. One such antiviral drug is the nucleoside analogue GS-441524, which is the active form of the prodrug remdesivir [22]. Recently published studies have demonstrated promising results with this antiviral drug [23,24].

However, access to these antiviral therapies for prescribers is currently problematic, as they have not yet obtained registration for veterinary use, particularly in mainland China [25,26]. As a result, treating FIP remains challenging.

Some studies in mainland China have also delved into viral genes at the molecular level [27]. Nevertheless, systematic prevalence studies in mainland China are conspicuously absent [28,29,30]. Additionally, the incidence of FIP exhibits significant variability across regions and countries [29]. FIP stands out as one of the most significant infectious diseases and leading causes of death in cats, particularly affecting young cats under two years of age, who are especially vulnerable. Research suggests that approximately 0.3% to 1.4% of feline deaths at veterinary institutions are attributable to FIP. Given the absence of registered antiviral drugs to treat FIP in mainland China, it has become crucial to understand the prevalence of FIP in this region and to identify potential risk factors. In this context, we present a meta-analysis aiming to estimate the prevalence of FIP in mainland China and assess potential risk factors, including sampling year, age, gender, breed, and season. The objective was to facilitate the formulation of effective strategies for the prevention and control of FIP in mainland China.

## 2. Materials and Methods

### 2.1. Data Sources and Retrieval Strategies

The study was conducted according to the Preferred Reporting Items for Systematic Reviews and Meta-Analyses (PRISMA) guidelines. The PRISMA checklist was used to ensure the inclusion of all relevant information in the analysis (Appendix A). PubMed, Web of Science, CNKI, Wanfang, and Science Direct databases were searched and the languages were limited to Chinese and English. In addition, there were four sets of data from pet clinics. The retrieval interval was from 1 January 2008 to 20 December 2023 and the references included in the study were manually retrieved. The retrieval strategy used a combination of subject words and free words. The data were from the literature and patents. The Chinese search formula was “FIP” or “FIPV” and “epidemiology” or “incidence” or “prevalence”, while the English search formula was “FIP” or “FIP virus”, and “epidemiology” or “incidence” or “prevalence”.

### 2.2. Inclusion Criteria

The data were included if they met the following conditions: (1) the studies used cats as the research subjects; (2) the studies reported the prevalence in cats; (3) the investigation locations were in mainland China; (4) the test method and process were clearly stated; and (5) the studies were written in English or Chinese. If the results were inconsistent, they were resolved by a third party or through negotiation and discussion. Articles that did not meet the above criteria were excluded.

### 2.3. Data Extraction and Quality Evaluation

Randomized controlled trials (RCTs) are widely considered the gold standard for establishing causal associations in clinical research. Although this non-randomized controlled study (a cross-sectional study) differs from a randomized clinical trial, its well-designed systematic evaluation method can provide valuable evidence for evidence-based medicine. Utilizing the Cochrane quality assessment tool, combined with an enhanced critical appraisal tool (High-Quality Item Rating Scale) developed by Munn et al. [31], we adapted the methodology for systematic evaluation in this study and assessed the risk of bias for the included studies [31]. Data extraction and recording were independently performed by three trained researchers. We then extracted the following information from each of the included studies: first author, sampling time, sampling location, total number of cats, numbers of FIP-positive cats, study design, age and sex of the animals, and detection method. Microsoft Excel 2017 was used for data management. RevMan 5.3 was used for quality assessment in terms of bias [32]. The following six items were examined and given a score based on a simple scale: 2 for “yes”, 1 for “unsure”, or 0 for “no”. (1) Were the research objectives/problems clearly described and stated? (2) Were the characteristics and location of the experimental animals clarified? (3) Was the sampling method described in detail? (4) Were the virus detection methods clearly noted? (5) Were the survey results subjectively influenced? Studies with a total quality evaluation score ≤ 6 were not included in the statistical analysis.

### 2.4. Statistical Analyses

The Stata software (version 15) was used for the meta-analysis, and the double-arcsine transformation (PFT) method was used to bring the data closer to a normal distribution [30,33]. Each study’s proportion estimate typically undergoes a specific transformation to achieve a better approximation to the normal distribution [30], aligning with the assumptions of conventional meta-analysis models. Subsequently, a meta-analysis was conducted on the transformed scale to assess the pooled prevalence [30]. The PFT conversion formulas are as follows:t = asin (sqrt (r/(n + 1)) + asin (sqrt (r + 1)/(n + 1)))
set = sqrt (1/(n + 1))
p = sin (t/2)^2^
where t is the detection rate after conversion, n is the total number of samples, r is the number of positive samples, set is the standard error, and p is the final detection rate.

Heterogeneity among the studies was estimated using the I^2^ test, following which the effect model was selected [34]. A random-effects model was selected if significant heterogeneity among studies was observed (*p* < 0.1 and I^2^ > 50%), according to the Cochrane handbook. The source of heterogeneity was analyzed through a meta-regression [35]. Otherwise, a fixed-effects model was used [36]. All effective quantities were expressed as 95% confidence intervals (CI), and *p* < 0.05 defined statistical significance.

### 2.5. Bias, Sensitivity Tests, and Subgroup Meta-Analysis

A funnel plot was employed to evaluate the presence of publication bias in the included studies. There was notable asymmetry in the funnel plot (with studies represented by dots), which suggested significant publication bias among the included studies. Egger’s test is commonly used to assess potential publication bias in a meta-analysis via funnel plot asymmetry. *p* ≥ 0.05 indicates that the risk of publication bias is small, while *p* < 0.05 indicates possible publication bias [37,38]. A sensitivity analysis was performed to assess the consistency and stability of our meta-analysis through systematically excluding one study at a time and recalculating the combined FIP risk [38]. Subgroup and meta-regression analyses were performed to evaluate the potential sources of heterogeneity and the factors that caused the heterogeneity. When analyzing the total prevalence of FIP, subgroups were analyzed by season, age, gender, and breed.

## 3. Results

### 3.1. Studies Included

According to the retrieval strategy applied to various databases, a total of 890 relevant articles were identified. Following the literature screening process outlined in the Cochrane manual, which involves assessing titles, abstracts, content, and exclusion criteria, the final selection included 21 articles [27,39,40,41,42,43,44,45,46,47,48,49,50,51,52,53,54,55,56,57,58]. These comprised 17 Chinese-language articles and 4 English-language articles for conducting a meta-analysis on the prevalence of FIP in mainland China, with a total sample size of 181,014 cases. All 21 studies were ultimately included in our meta-analysis. The following data were extracted: the first author, publication year of the journal, survey area, no. positive, no. examined, and detection method (Table 1). Figure 1 provides a detailed overview of the selection process and results.

### 3.2. Quality Assessment and Data Extraction

The quality of each study was evaluated, and the corresponding scores are depicted in Figure 2. Green represents 2 points, indicating low risk; yellow represents 1 point, indicating uncertainty; and red represents 0 points, indicating high risk. Based on the evaluation of five quality assessment criteria, with a maximum score of 10 points, the scores for the included studies ranged from 7 to 10 points. The overall average quality score was 8.86 ± 1.06, with a median score of 9, as depicted in Figure 2.

### 3.3. Heterogeneity Analyses

Double arcsine conversion was applied to the FIP prevalence data reported in the 21 studies to estimate the prevalence of FIP and 95% CI. Through a heterogeneity test for 21 articles included in the final analysis, an I^2^ value of 95.2% was obtained, indicating significant heterogeneity (I^2^ > 50%). Combining the incidence rates from each study, the analysis was performed using a random-effects model. The results reveal that the incidence rate of FIP in the Chinese region is 2% (95% CI: 1–2%), as illustrated in Figure 3.

### 3.4. Publication Bias and Sensitivity Analysis Bias

Applying Egger’s test to analyze potential publication bias yielded a result of *p* = 0.001 < 0.05, indicating a relatively high risk of publication bias, as shown in Figure 4. Given the substantial heterogeneity observed in the meta-analysis of FIP incidence rates, a sensitivity analysis was performed on the included literature. Through the systematic removal of individual studies for sensitivity analysis [59], the results indicated that the effect sizes consistently fell within the 95% confidence interval of the final effect size. This demonstrates the good stability and robustness of the results, as depicted in Figure 5.

### 3.5. Subgroup Meta-Analysis

A subgroup analysis was conducted to identify sources of heterogeneity, the results of which are presented in Table 2 and Figure 6. Due to the significantly high heterogeneity between subgroups in most studies, pooled prevalence estimates for each subgroup were calculated using the random-effects Dersimonian–Laird model.

Breed subgroup analyses revealed substantial differences in FIPV infection prevalence across genders (*p* = 0.000). Garfield exhibited the highest prevalence at 4% (95% CI, −1–8%), followed by British Shorthair at 3% (95% CI, 2–3%), Idyllic Cat at 3% (95% CI, 1–5%), Muppet at 2% (95% CI, 1–4%), and American Shorthair and Other with the lowest prevalence at 1% (95% CI, 0–2%) (Figure 6A). The age subgroup analysis demonstrated that the pooled prevalence of FIPV was 2% (95% CI, 1–3%) for cats over two years old; however, it significantly increased to 15% (95% CI, 8–22%) for kittens aged two years and below, with an extremely significant difference (*p* = 0.000) observed between these two age groups (Figure 6B). There were significant differences (*p* = 0.000) in prevalence by gender, with the highest prevalence of FIPV in entire males at 10.0% (95% CI, 9–11%), followed by entire females at 9.0% (95% CI, 7–11%), spayed females at 3% (95% CI, 2–4%), and the lowest in castrated males at 2% (95% CI, 1–3%) (Figure 6C). No significant differences (*p* = 0.480) in prevalence were observed for season (Figure 6D).

## 4. Discussion

FCoV is globally distributed and prevalent in most cat populations, exhibiting particularly high contagion rates in multi-cat environments such as breeding catteries and shelter/rescue facilities [2]. In crowded living conditions where the sharing of litter boxes and feeding bowls is common, FcoV infections are widespread [3]. We conducted a meta-analysis based on 21 Chinese- and English-language articles, revealing a 2% prevalence rate of FIP in mainland China. These findings suggest that FIP is prevalent, to some extent, in mainland China.

### 4.1. Breed

The findings of this study revealed that, in pet clinics in mainland China, purebred cats such as Garfield, Idyllic, and British Shorthair have a higher infection rate of FIP. This may be related to the number of cat breeds visiting hospitals in China. According to the 2023 “China Feline Diagnosis and Treatment White Paper”, the proportion of cat breeds in pet hospitals are 33.4% for British Shorthair and 27.1% for Idyllic [2]. It may also be related to the examined small sample size (530) of Garfield in this study (Table 2).

In Loretta D. Pesteanu-Somogyi’s study investigating feline accessions at the North Carolina State University College of Veterinary Medicine over the 16-year period from 1986 to 2002, of the 11,535 cats of known breed that were examined, the prevalence of suspected or confirmed FIP in the mixed-breed cat population was 0.35% versus 1.3% in the purebred cat population [60]. The more purebred a cat is, the higher the degree of genetic overlap between its parents, leading to an increased likelihood of hereditary diseases and a decrease in resistance. In contrast, hybrid cats generally exhibit stronger resistance and survivability, compared to purebred cats. Although breeds such as British Shorthair and Ragdoll have a higher incidence of the disease, there are also cat breeds that are less susceptible to FIP. In the study discussed above, the final diagnosis in all cases was determined by the attending clinician. Interestingly, the prevalence of the disease was found to be nil in 23 cat breeds. However, there are conflicting reports stating that breed is not related to FIP [61]. This may be related to factors such as the cat rearing environment, survey scale, survey time, and cat preference involved in the study [60,62]. For example, male cats are more common as pets than female cats, resulting in a higher proportion of positive cases being observed in male cats in epidemic investigations [40]. In addition, in the purebred cat population, the prevalence of FIP can vary greatly between regions and countries [29,60,63,64,65]. More research is needed to investigate the correlation between breeds and FIP in cats.

### 4.2. Age Prevalence of FIP

All age groups of cats are susceptible to FIP. This study in mainland China revealed a significantly higher incidence rate in cats aged two years or below compared to those aged two years and above (15% vs. 2%). This observation indicated that kittens have a greater risk of contracting FIP than adult cats [62,64,66]. Other studies have also reported a significant correlation between age and FIP, with kittens (under two years of age) being more susceptible [61,67].

### 4.3. Gender

The results indicated that the prevalence of FIP in intact cats (male or female) was significantly higher than that in castrated males or spayed females (Table 2). Similar research results have also been found in male cats [61,62], as male cats were more prone to FIP [68]. This may be attributed to sex hormones, especially androgens, exerting a negative effect on the immune system, thereby elevating the risk of virus proliferation and mutation [69] and resulting in a higher prevalence in male cats. The elevated activity level of male cats, driven partly by the need to compete for mating rights and protect their territory and resources, often led to intense fights, increasing the likelihood of virus exposure and making them more prone to developing FIP. The multifactorial nature of FIP susceptibility suggests that, while sex hormones may play a role, other factors also contribute to the overall risk of infection.

### 4.4. Seasonal Prevalence of FIP

Outdoor activities among cats, as well as heightened interactions with various cats, including strays and free-roaming cats, during warmer temperatures, significantly elevate the risk of contact infections [39,54,57]. Summer and autumn, as the typical seasons for kittens to be weaned, introduce sudden changes to their reproductive and living environment. This can potentially result in a decrease in their immune system’s resilience against viruses [25]. In our study, there were no differences in the seasonal prevalence of FIPV infections in mainland China. Due to limited similar studies in veterinary medicine, evaluating the effect of seasonality on infection type remains challenging [70]. To gain a comprehensive understanding, additional long-term and large-scale seasonal studies on feline FIP are essential. Nonetheless, we recommend reinforcing cat management and enhancing immunity during the summer and autumn cat mating season to prevent FIPV infections. Strict control over the activity range of FIP-positive cats could help to minimize the spread of the virus.

### 4.5. Early Weaning and Stress

Early weaning and isolation of cats that test positive for coronavirus antibodies are effective measures for the prevention and control of FIP. A pregnant female cat, through mating, may vertically transmit feline enteric coronavirus (FECV) or feline Coronavirus (FcoV) to her offspring. It is possible that kittens already carry the respective virus at birth, and the disease may manifest when conditions are conducive. An early weaning at 4 to 6 weeks of age, as a protocol for the prevention of FCoV infection in kittens, has been proposed [71,72]. Due to the crowded housing situation in mainland China, it is difficult to provide enough space for the isolation of mother cats, so early weaning is not as simple as it might seem. Therefore, in facilities with large numbers of cats, it can be very difficult to eliminate feline coronavirus and there will be more risk of the development of FIP.

Stress also plays a large role in whether an FCoV-infected cat develops FIP. Cats that have recently undergone a stressful event are also more likely to develop FIP [2]. Stressors such as moving to a new environment, cat density, or surgery may increase the risk of an individual developing FIP [73]. Higher incidence and outbreaks are expected in stressful environments. Given the crucial role of stress in FIP development [19], it is advisable to minimize unnecessary stressors, such as rehoming and elective surgery, to potentially benefit the overall well-being of the cat.

### 4.6. Detection

The definitive diagnosis of FIP traditionally relies on histopathological examination, with open abdominal exploration and autopsy serving as the ultimate methods. The detection methods involved in this study included clinical examinations, hematological testing, radiography (chest and abdominal), abdominal and chest ultrasound examination, effusion examination (pleural effusion, ascites), RT-PCR, and histopathological examinations (Table 1).

Clinical testing included a thorough review of history, signalment, and clinical presentation. Clinical signs include anorexia, lethargy, weight loss, and pyrexia. Cats with ocular FIP may also show signs of conjunctivitis, characterized by redness, discharge, and swelling of the conjunctiva. However, this symptom is non-specific as Mycobacterium, Toxoplasma gondii, FIV, and FeLV all cause similar intraocular lesions, and may also cause retinal hemorrhage in cats with hypertension. Hematological abnormalities are very common in cats with FIP.

Compared to traditional PCR detection methods, RT-PCR is faster and can perform virus quantification, which is also a commonly used rapid PCR detection method in clinical practice. It is well-known that cats with FIP exhibit much higher viral loads than healthy FECV-infected cats [74,75]. A positive RT-PCR result with a high viral load is at least suggestive of FIP [3]. The source samples are also crucial for the diagnosis of FIP. Clinical diagnosis of FIP requires a comprehensive evaluation based on sample results. It is now accepted that antibody tests cannot differentiate between antibodies against FECV and FIPV and, so, high antibody titers in blood are not a specific indicator of FIP [76]. In addition, for cases that are difficult to diagnose, open abdominal exploration is the most effective diagnostic approach. Partially affected tissue can also be cut out and sent to a molecular pathology diagnostic laboratory for further diagnosis through tissue fluorescence staining and immunohistochemistry. In terms of clinical symptoms, FIP closely resembles those of various other diseases such as jaundice and hypoalbuminemia, leading to potential confusion during diagnosis. Therefore, it is essential to thoroughly investigate the etiology, medical history, symptoms, signs, examination results, and responses to drug treatments. A comprehensive analysis is necessary to make an accurate diagnosis.

### 4.7. Infection of Felidae

In addition to domestic cats, FIPV may also infect some wild felids globally, such as African lions, mountain lions, leopards, cheetahs, jaguars, lynx, servals, caracals, European wildcats, sand cats, Pallas’ cats, and nearly all other catamounts [72,77,78,79,80,81]. Due to the close antigenicity relationship between coronaviruses from different animal species (such as CCV and TGEV) and coronaviruses originating from animals in close contact with humans, people are concerned that FCoV may pose a danger to humans. However, there is no indication that people will be infected with FCoV [82]. Nevertheless, research on the pathogenesis, diagnosis, prevention, and treatment of FCoV has been helpful in the treatment of human coronavirus infection [82].

### 4.8. Prevention and Treatment of FIP

There is still no effective FCoV vaccine, which continues to limit the prevention and treatment of the disease caused by this virus. Additionally, there is an ongoing need to enhance the accuracy, sensitivity, and specificity of antemortem FIP diagnosis in cats.

The meta-analysis presented in this study, based on 21 articles reporting on FIP in mainland China, had a few limitations: (1) The majority of collected samples originated from urban, domesticated cats, with few cases reported in free-roaming and stray cats in rural areas. This bias may impact the generalizability of the findings to the broader feline population. (2) This study only included the published literature; unpublished documents were not considered, potentially introducing publication bias. (3) Feline Panleukopenia Virus (FPV) is a pathogen included in the core vaccination schedule for cats. In this study, the recent vaccination status of the included cats was unknown. Therefore, individuals recently vaccinated against the virus might exhibit false-positive results.

## 5. Conclusions

FIP is prevalent, to some extent, in mainland China, and the overall combined prevalence of FIP was found to be 2%. The incidence of FIP is influenced by age and gender in mainland China. We recommend enhancing cat immunity, especially focusing on health management in kittens and intact cats, and continuously monitoring FIPV to effectively reduce and control the prevalence of FIP in mainland China.

## Figures and Tables

**Figure 1 animals-14-01220-f001:**
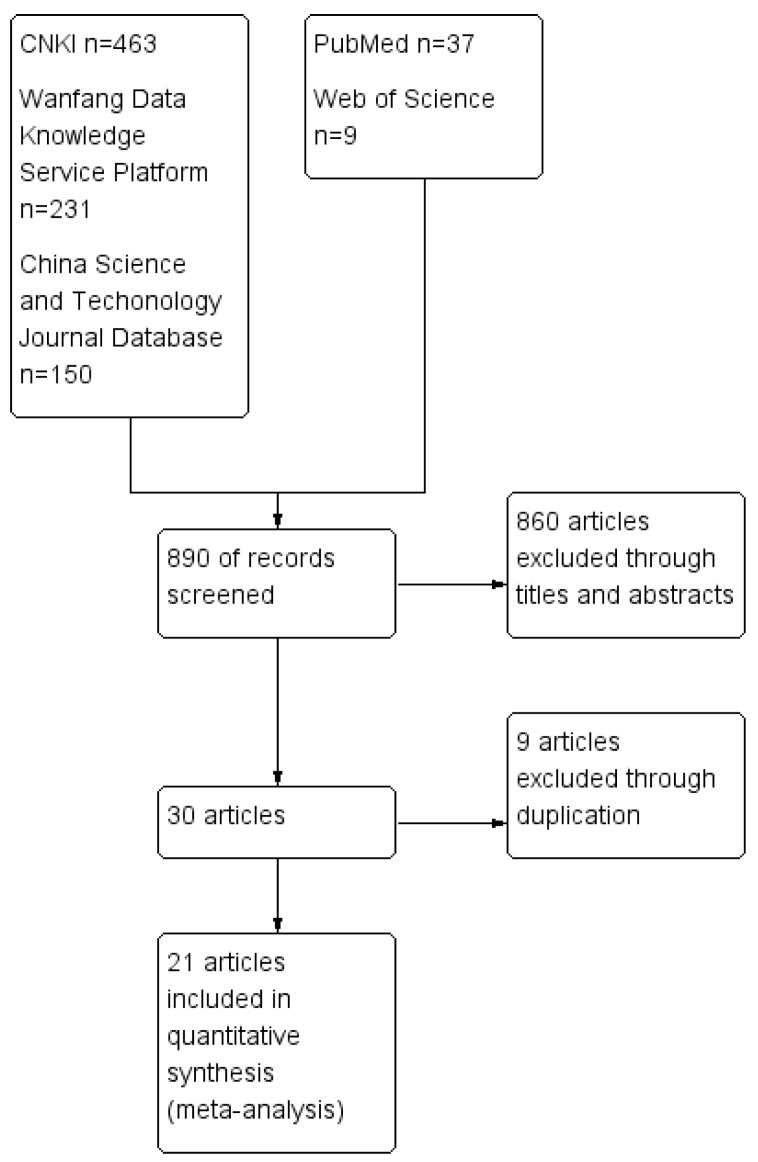
Flow diagram of reference screening on FIP.

**Figure 2 animals-14-01220-f002:**
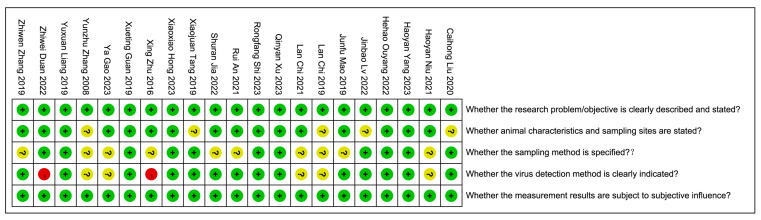
Quality assessment of studies screening on FIP [27,38,39,40,41,42,43,44,45,46,47,48,49,50,51,52,53,54,55,56,57].

**Figure 3 animals-14-01220-f003:**
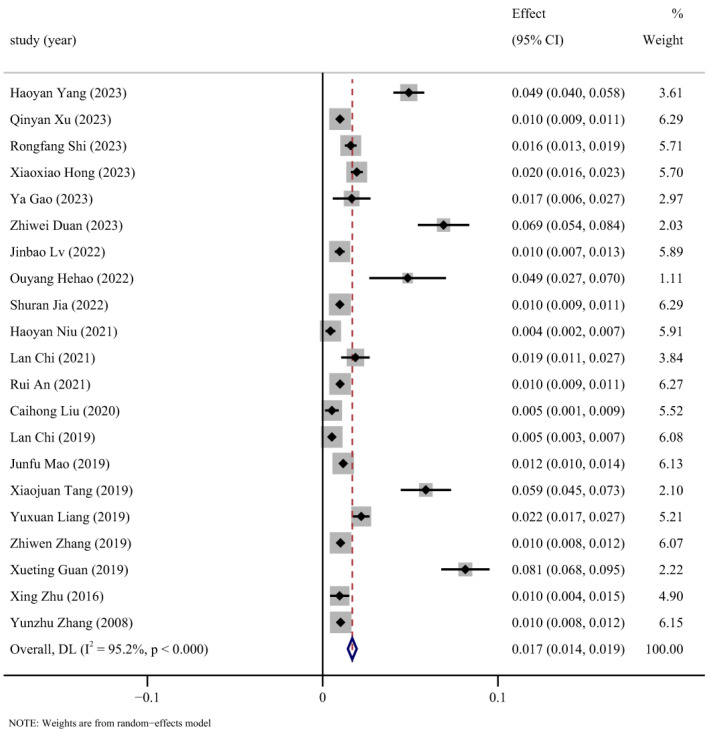
Summary forest plot of FIP incidence with random-effects analyses [27,38,39,40,41,42,43,44,45,46,47,48,49,50,51,52,53,54,55,56,57]. ES: Effect size (positive proportion).

**Figure 4 animals-14-01220-f004:**
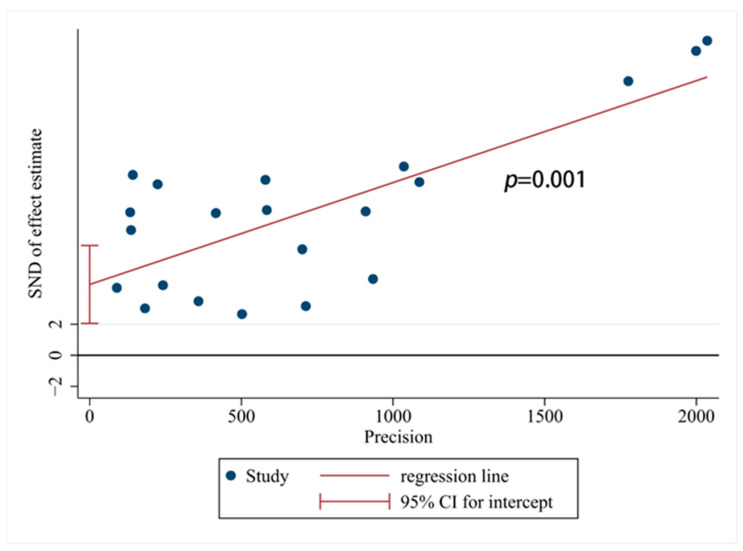
Egger test of publication bias among FIP studies. SND: Standard normal deviation; Precision: Inverse of the variance.

**Figure 5 animals-14-01220-f005:**
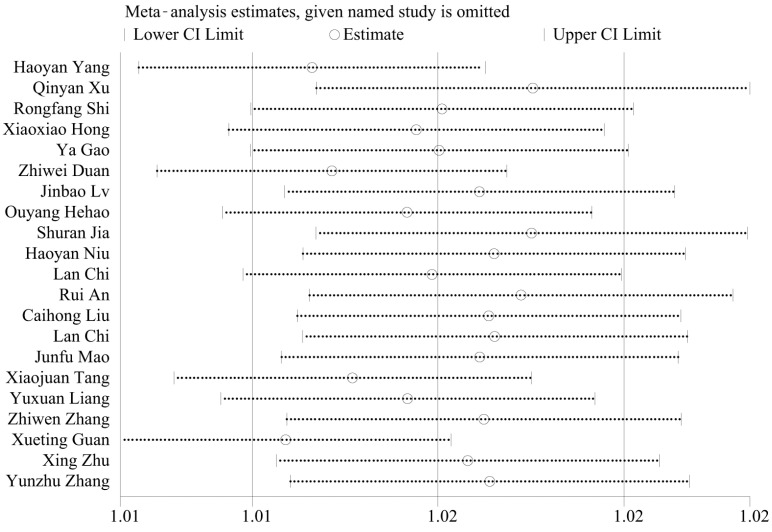
Sensitivity analysis of FIP incidence [27,38,39,40,41,42,43,44,45,46,47,48,49,50,51,52,53,54,55,56,57].

**Figure 6 animals-14-01220-f006:**
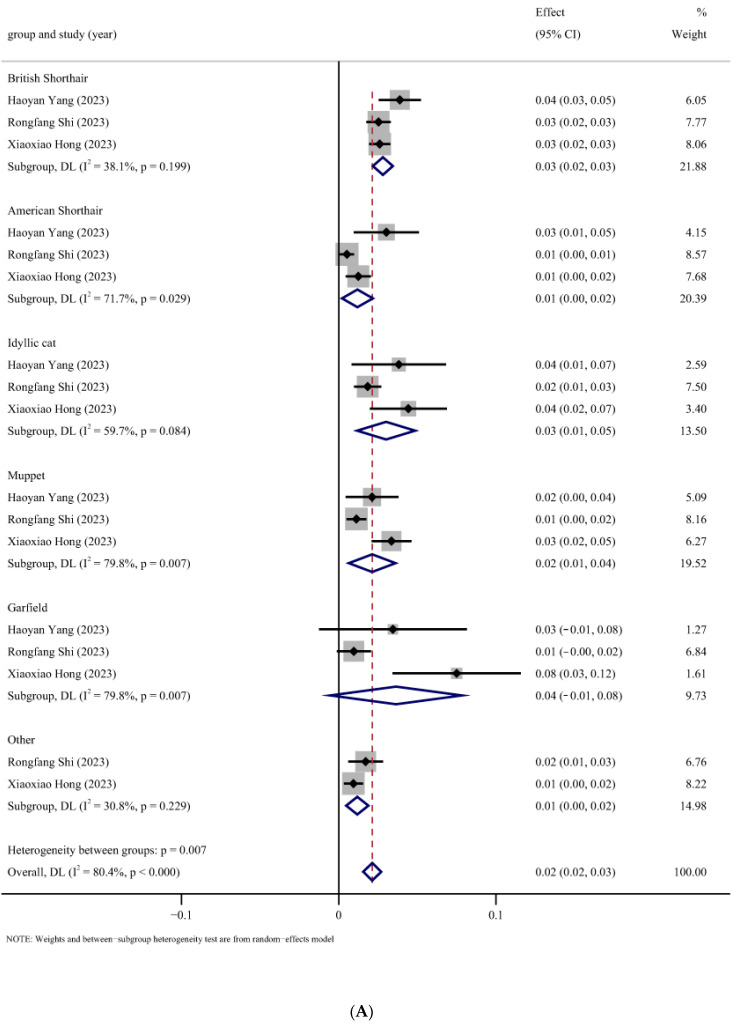
Forest plot of the FIP estimated incidence subgroup with random-effects analyses. (**A**) The subgroup analysis of different breeds [43,50,54]; (**B**) the subgroup analysis of different ages [41,43,54]; (**C**) the subgroup analysis by gender [38,56]; and (**D**) the subgroup analysis by season [38,50,52,56]. ES: Effect size (positive proportion).

**Table 1 animals-14-01220-t001:** Baseline data of the included literature.

Author	Publication Year	Survey Area	No. Positive	No. Examined	Detection Method
Haoyan Yang [55]	2023	Xi’an	115	2339	RT-PCR
Qinyan Xu [54]	2023	Jinan	406	40,820	Hematology, imaging examination, immunohistochemistry techniques
Rongfang Shi [51]	2023	Shenzhen	86	5375	PCR
Xiaoxiao Hong [44]	2023	Qingdao	125	6410	RT-PCR
Ya Gao [43]	2023	Huhehot	9	543	RT-PCR
Zhiwei Duan [42]	2023	Lanzhou	79	1145	Imaging examination
Jinbao Lv [47]	2022	Beijing	46	4732	qPCR
Ouyang Hehao [27]	2022	Central China	18	371	RT-PCR
Shuran Jia [45]	2022	Shenyang	380	38,775	RT-PCR
Haoyan Niu [49]	2021	Nanjing	10	2249	RT-PCR
Lan Chi [40]	2021	Xuzhou	20	1068	Unmarked
Rui An [39]	2021	Beijing-Tianjin-Hebei Region	308	31,001	RT-PCR
Caihong Liu [52]	2020	Multiple regions	7	1326	RT-PCR
Lan Chi [41]	2019	Shanghai	24	4564	Unmarked
Junfu Mao [48]	2019	Beijing	146	12,439	Hematology, imaging examination
Xiaojuan Tang [53]	2019	Wuhan	61	1036	RT-PCR
Xueting Guan [50]	2019	Harbin	124	1523	RT-PCR
Yuxuan Liang [46]	2019	Nanjing	82	3729	Hematological examination, PCR
Zhiwen Zhang [57]	2019	Dalian	85	8341	RT-PCR
Xing Zhu [58]	2016	Guiyang	12	1236	Unmarked
Yunzhu Zhang [56]	2008	Beijing	123	11,992	Unmarked

**Table 2 animals-14-01220-t002:** Frequency of FIP-positive cats in cat clinic populations and subgroup analysis.

Factor	No. of Studies	No. Positive	No. Examined	Prevalence *		Heterogeneity			*p*
				Estimates	(95% CI)	PQ	I^2^ (%)	Q(χ^2^)	
Age	3								
2 years old or below		273	1954	0.15	0.08, 0.22	<0.000	95.6%	693.86	0.000
Over 2 years old		42	2522	0.02	0.01, 0.03	0.046	67.5%	8.84	
Gender	2								
Castrated male		40	1958	0.02	0.01, 0.03	0.461	0.0%	0.54	0.000
Entire male		221	2241	0.10	0.09, 0.11	0.636	0.0%	0.22	
Spayed female		23	742	0.03	0.02, 0.04	0.770	0.0%	0.09	
Entire female		109	1197	0.09	0.07, 0.11	0.287	11.9%	1.14	
Breed	3								
Muppet		44	2129	0.02	0.01, 0.04	0.007	79.8%	159.59	0.007
Garfield		17	530	0.04	−0.01, 0.08	0.007	79.8%	81.84	
British Shorthair		128	4564	0.03	0.02, 0.03	0.199	38.1%	756.75	
American Shorthair		22	1950	0.01	0.00, 0.02	0.029	71.7%	35.43	
Idyllic Cat		35	1353	0.03	0.01, 0.05	0.084	59.7%	87.15	
Other **		18	1489	0.01	0.00, 0.02	0.229	30.8%	1.45	
Season	4								
Spring		82	1958	0.07	0.02, 0.11	<0.000	94.0%	307.63	0.480
Summer		224	3276	0.08	0.02, 0.14	<0.000	97.3%	340.58	
Autumn		177	5646	0.04	0.02, 0.06	<0.000	93.6%	315.28	
Winter		57	1557	0.05	0.01, 0.10	<0.000	93.6%	186.26	

*: Confirmed (positive) proportion in clinics. **: Other breeds of cats (Siamese cats, Manx cats, Maine Coon cats, and other breeds), as well as cats of unknown breeds.

## Data Availability

Data are contained within the article.

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
