# Peer review of "Prevalence and Risk Factors Associated with Feline Infectious Peritonitis (FIP) in Mainland China between 2008 and 2023: A Systematic Review and Meta-Analysis"

_animals, 2024, doi:10.3390/ani14081220_

Round 1
Reviewer 1 Report
Comments and Suggestions for Authors
Prevalence and risk factors associated with Feline infectious 2 peritonitis (FIP) from mainland China between 2009-2023: a 3 systematic review and meta-analysis
Please consider that references should be indicated into parentheses or square brackets and separated by colons.
Regarding the rate of FIP in the Chinese region, It seems to me that its 2% result (95% CI: 1%-2%), with a confidence interval of almost equal. What can you discuss about tis?
Figure 6. can´t be read well
Line 220. Is there a possibility that these breeds are proportionally more abundant in China than the rest and are, somehow more represented in the studies considered in this manuscript? Is there a census that indicates which breeds are most abundant in China?
Line 301. abdominal
Line 329. Consider to use: Nevertheless, Nonetheless, Although….instead to repeat However.
Line 348. In this meta-analysis, the incidence of FIP seems to be influenced by age, gender, and breed of cats in China. (Since some papers indicate that breed is not associated with FIP: Purebred cats were previously considered to be more prone to FIP. However, a growing number of studies, including this study, have shown that the proportion of purebred cats was not excessive when compared with the clinic population (doi: 10.1038/s41598-021-84754-0).
Author Response
Thank you very much for taking the time to review this manuscript. Please find the detailed response below and the corresponding revisions/corrections in the resubmitted document.

Reviewer 2 Report
Comments and Suggestions for Authors
In this study, the author comprehensively reviewed literature on the prevalence of FIP in cats in mainland China from January 1, 2008, to December 20, 2023, retrieved from five databases: CNKI, Wanfang, Pub-Med, Web of Science, and ScienceDirect. The selected 21 studies, with a total of 181 014 samples, underwent a rigorous meta-analysis after assessing literature quality. The results revealed a 2% prevalence of FIP (95% CI: 1%-2%) through the random-effects model, showing considerable heterogeneity (I2=95.2%). The subsequent subgroup analysis unveiled that age, gender, and breed of cats are significant risk factors for FIP infection in mainland China. However, the readability of the manuscript can also be greatly improved. Through editing and some modifications, I think this manuscript will be more suitable for publication.
1. The Abstract section should preferably be a summary of an article, without the need for paragraphs. In addition, the Abstract only provides a brief description of the results, lacking conclusions and insufficient to highlight the research significance of the article.
2. Line 27, “181014” samples, please use international standard writing for the numbers in the article.
3. Introduction section, There is too little description of research on FIP internationally, and this section should be added.
4. P in P<0.1 or P<0.05 should be in italics, please check the entire article for consistent formatting.
5. Materials and methods need more references to support the validation.
6. References should be modified according to the requirements of the journal, rather than randomly inserting numbers into the main text of the article.
7. All Figures in the manuscript are distorted, and the author should provide Figures with higher and clearer pixels
8. Language should be improved to better the understanding.
Comments on the Quality of English Language
Moderate editing of English language required
Author Response
Thank you very much for taking the time to review this manuscript. Please find the detailed responses below and the corresponding revisions/corrections highlighted changes in the re-submitted files.

Reviewer 3 Report
Comments and Suggestions for Authors
References should go between brackets, it creates confussion.
You should give a brief structure description of the disease with symptons, transmission, diagnosis, treatment...but not only giving a sentence of the clinic.
You should describe the importance of the disease to give value to your study, why did you choose this and not other virus...
You should be concise writing the aim of the study. Before that or even in materials and methods you can say the factors included.
Use some kind of order in the tables and figures (the same) to clarify the information for the reader, i suggest by the name of the first author or maybe by the year of publication of the research.
In table 2 change the order, writing first the most important factors and at the end the no-significance ones.
Figure 6: it is difficult to read the information. Take the same structure, put in the same order all of them and if you have some articles with no reference to the gender (for example) put 0 instead of omiting it.
You do not mention any research as in line 229. Use the same structure to mention some other researches and write the year of publication. But it is better if you take this method instead of writing a sentence with no connection after other one. You should create a fluent text in order to prevent the lost of the meaning of the research.
I do not understand why are you talking about US and Taiwan in point 4.2 if your study is focused in China and there is no other comparative in the rest of points.
In point 4.3 change the redaction. you have some research that say there is no important the sex but other which say yes...make connection. The same for point 4.5
In point 4.6 you talk about histopathological examination, you should mention at the beginning if it is in death or alive animals...explain a little bit
Change redaction about symptons, if they are not only for PIF like...On the other hand, all these symptons can have different aethiological origin or similar.
Line 320, you could use different words as affected instead of diseased and indicate from where you are making the biopsy.
in point 4.8 you talk abour 16 articles. you should indicate that you are talking about 16 from the 21 you have already selected.
Author Response

(The authors gave the same response as above.)

Reviewer 4 Report
Comments and Suggestions for Authors
Feline infectious peritonitis (FIP) is an important widespread and susceptible disease of domestic cats. It occurs worldwide in cats of all ages, but the disease is most common in young cats less than two years of age. FIP is associated with a viral infection called feline coronavirus. There are many different strains of feline coronavirus, which differ in their ability to cause disease. The aim of the present study was evaluate the prevalence of FIP infection in cats in mainland China and associated risk factors, although a comprehensive review of literature. In my opinion, the study has very limitated information related the area examined, having a major interest for Chinese people. The inclusion of Chinese literature not available for non Chinese scientific readers, and the topic treated is very limited in scientific soundness and very poor interest for the readers. Also, the text is very poor comprehensive for the presence of numbers apparently not related of the references. The quality of the figures is very low.
|
Extensive editing of English language required.
Author Response

(The authors gave the same response as above.)

Round 2
Reviewer 3 Report
Comments and Suggestions for Authors
You have changed and responded all the comments we have done. I can see the differences between the old version and this one. It is a great job. Well done.
Reviewer 4 Report
Comments and Suggestions for Authors
Thank you for your efforts.